# Mechanisms of Andreev reflection in quantum Hall graphene

Antonio L. R. Manesco[1,2*], Ian Matthias Flór [2],
Chun-Xiao Liu[2,3] and Anton R. Akhmerov[2]

**1** Computational Materials Science Group (ComputEEL), Escola de Engenharia de Lorena,
Universidade de São Paulo (EEL-USP), Materials Engineering Department, Lorena - SP, Brazil
**2** Kavli Institute of Nanoscience, Delft University of Technology,
Delft 2600 GA, The Netherlands
**3** Qutech, Delft University of Technology,
Delft 2600 GA, The Netherlands

★ am@antoniomanesco.org

## Abstract

We simulate a hybrid superconductor-graphene device in the quantum Hall regime to identify the origin of downstream resistance oscillations in a recent experiment [Zhao *et al.* Nature Physics 16, (2020)]. In addition to the previously studied Mach-Zehnder interference between the valley-polarized edge states, we consider disorder-induced scattering, and the appearance of the counter-propagating states generated by the interface density mismatch. Comparing our results with the experiment, we conclude that the observed oscillations are induced by the interfacial disorder, and that lattice-matched superconductors are necessary to observe the alternative ballistic effects.

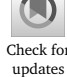
# 1 Introduction

Already since the early years of graphene, researchers were able to fabricate and measure high-quality graphene–superconductor devices [1, 2]. The ease of fabrication inspired a plethora of works examining tunneling spectroscopy [3, 4], Josephson effect [2, 5–7], multiple Andreev reflections [8], imaging Andreev scattering, [9] quantum phase transitions [10, 11], reflectionless tunneling [12], microwave circuits [13], and bolometer devices [14] as well as other physical phenomena. Because quantum Hall effect in graphene manifests already at relatively low magnetic fields below 1T, graphene is also uniquely fit to combine quantum Hall physics with superconductivity and Andreev reflection [15–18].

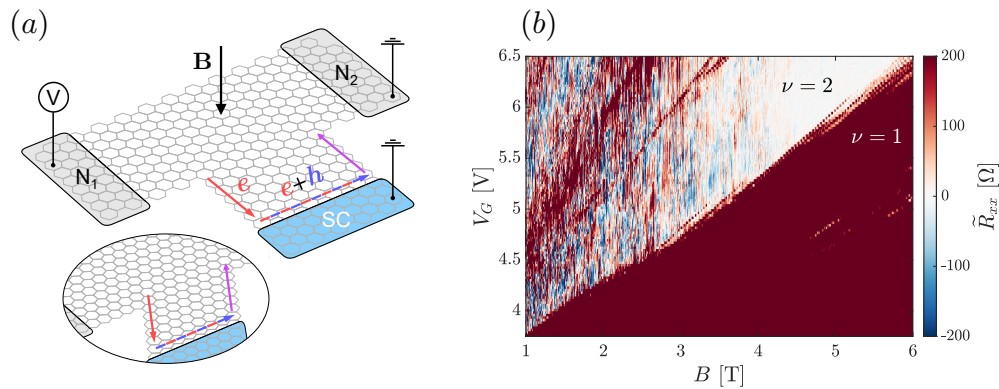

Figure 1: (a) An example device for probing nonlocal Andreev reflection in graphene. The device has 3-terminals: two normal leads (grey) and a superconducting lead (blue). Both graphene edges connected with the superconducting lead are zigzag, which mimics the valley polarization of a generic graphene boundary. The interface can be either armchair (main figure) or zigzag (inset). (b) The measured downstream resistance $\widetilde{R}_{xx}$ as a function of the magnetic field $B$ and the gate voltage $V_G$ in a multi-terminal setup (courtesy of Zhao *et al.* [16]). The separation between filling factors $\nu = 1$ and $\nu = 2$ is highlighted.

A recent work observed a strongly fluctuating downstream resistance in a multiterminal device as a function of the magnetic field and the gate voltage, shown in Fig 1(b). The authors have interpreted the data as chiral Andreev edge states interference, supported by tight-binding calculations [16]. This finding contradicts the idealized theory [19] that predicts that Andreev conductance of graphene in the lowest quantum Hall plateau should be constant. The constant conductance is a consequence of three factors combined. First, the boundary conditions imposed by generic graphene terminations ensure valley number conservation and also favor the population of one sublattice [20, 21]. Second, the lowest Landau level states are valley-sublattice locked [20, 22], resulting in valley-polarized quantum Hall edge states. Finally, at the interface with a superconductor, electron edge states hybridize with hole states with opposite valley isospin and the resulting chiral Andreev edge states generate a nonlocal transport signal. Hence, conductance depends solely on device geometry (see App. A for details) [19]. In particular, if the two edges connected to the superconductor are parallel, as depicted in Fig. 1(a), the conductance between the two normal leads equals

$$G_{xx} = -\frac{2e^2}{h},\tag{1}$$

where $e$ is the electron charge and $h$ is the Planck constant [19].

Our goal is to investigate Andreev states' interference and identify its possible origins.

We identify three mechanisms leading to deviations from constant conductance. The first option, proposed in Ref. [16], is the Andreev interference created by the lattice mismatch (Sec. 2). We find that most interface orientations lead to vanishing Andreev interference due to the suppressed intervalley coupling. Turning to another possibility, the short-range disorder (Sec. 3), we confirm that the large momentum transfer results in irregular oscillations between normal and Andreev reflections at any interface orientation. Finally, a sufficiently high Fermi level mismatch (Sec. 4) generates additional edge states and leads to interference through intravalley scattering.

All three mechanisms produce conductance fluctuations, albeit with different characteristics. Lattice and Fermi level mismatch at perfect interfaces generate a regular interference pattern in nonlocal conductance due to the translational invariance of the Hamiltonian. An irregular interference pattern, similar to the experimental data, occurs only in the presence of a strong disorder. We compare the results with experimental data and discuss the relevance of our findings in Sec. 5.

## 2 Andreev interference in clean graphene quantum Hall devices

The NS interface hosts two co-propagating Andreev states (throughout the manuscript we treat spin as a trivial degeneracy). The linearized Hamiltonian of these two states is constrained by particle-hole symmetry and has the general form

$$H_{\text{eff}} = v(k\sigma_0 + k_0\sigma_z) + m_x\sigma_x + m_y\sigma_y , \tag{2}$$

where $k$ is the momentum along the interface, $v$ is the Fermi velocity of the propagating chiral modes, $\sigma_0$ is the identity matrix, and $\sigma_{i=x,y,z}$ are Pauli matrices. We choose the basis in which the valley operator in graphene is $\sigma_z$, so that the two couplings $m_x$ and $m_y$ set the intervalley scattering strength, and $vk_0$ is the valley splitting. For illustration purposes we derive the Hamiltonian (B) starting from the continuum model of graphene in the App. B. Chiral edge states at a normal graphene boundary are in general valley-polarized [21], so that the Hamiltonian along a normal interface is diagonal in the valley basis. Therefore a negligible coupling $m_x$, $m_y \ll vk_0$ between the two chiral states preserves the valley polarization and consequently leads to either perfect or absent electron-hole conversion depending on the boundary conditions of the two normal edges [19]. A strong coupling $m_x$, $m_y \gtrsim vk_0$, on the other hand, results in Mach-Zehnder interference as a function of the magnetic field and gate voltage, as demonstrated by numerical simulations in Ref. [16].

Since valley isospin is preserved in graphene in absence of short range disorder, the intervalley scattering rate is bounded from above by the attempt rate in which quasiparticles in graphene hit the NS interface. This attempt rate is set by the low-energy scale $vk_F$, with $k_F$ the Fermi momentum in graphene, and therefore $m_x$, $m_y \lesssim vk_F$. Because the chiral states are composed out of states at the Fermi energy and have a high wave function weight in graphene, their momenta coincide with the projection of the valley momentum on the NS interface up to $\sim k_F$. Hence, we conclude that $k_0 \approx K\sin\theta$; where $\theta$ is the angle between the nearest armchair direction and the NS interface orientation, $K = 4\pi/3\sqrt{3}a$ is the magnitude of the valley momenta in graphene, and $a$ is the graphene lattice constant. Comparing these estimates of $vk_0$ and $m_x$, $m_y$ therefore predicts that for the Mach-Zehnder conductance oscillations to be visible, the NS boundary must be aligned with an armchair direction within an angle of $\lesssim k_F a \sim 1°$. This qualitative argument is confirmed by our tight-binding simulations as follows. First, in absence of valley splitting when the NS interface is along the armchair direction, we observe that the intervalley scattering is indeed small, as shown in Fig. 2 (a). Second, the chiral edges are valley-polarizated when the NS interface is oriented along the zigzag both

when the superconductor has a lattice mismatch [Fig. 2 (b)] and in presence of an atomically sharp electrostatic potential [Fig. 7 (a)].

To confirm the absence of interference at clean NS interfaces with generic orientation (small $m_x$, $m_y$), we perform tight-binding calculations using the Kwant package [23]. The Hamiltonian reads

$$\mathcal{H} = \sum_i \psi_i^\dagger (\Delta_i \tau_x - \mu_i \tau_z) \psi_i - \sum_{\langle i,j \rangle} \psi_i^\dagger (t_{ij} e^{i\tau_z \phi_{ij}} \tau_z) \psi_j, \tag{3}$$

where $\psi_i = (c_i, c_i^\dagger)^T$, $c_i^\dagger$ and $c_i$ are the electron creation and annihilation operators at the position $\mathbf{r}_i$, and $\langle i,j \rangle$ are all the pairs of nearest neighbor sites. We simulate the interface by using the following position dependence of the chemical potential $\mu_i$ and the superconducting pairing potential $\Delta_i$:

$$\mu_i = (\mu_{SC} - \mu_{QH}) f(\mathbf{r}_i) + \mu_{QH}, \quad \Delta_i = \Delta \Theta(x_i), \quad f(\mathbf{r}_i) = \frac{1}{2}\left[1 + \tanh\left(\frac{x_i}{\chi}\right)\right], \tag{4}$$

with $\mu_{QH}$ and $\mu_{SC}$ the onsite energies at the normal and the superconducting region. The Pauli matrices $\tau_i$ act on the electron-hole spinor components. The hopping energies $t_{ij} = t$ are constant in the honeycomb crystal structure, and equal to $t_{ij} = t/2$ in the square lattice that we use to simulate a lattice mismatch with the superconductor. The Peierls phase is:

$$\phi_{ij} = -\frac{\pi B}{\phi_0}(y_j - y_i)(x_j + x_i) \Theta\left(-\frac{x_i + x_j}{2}\right), \tag{5}$$

where $B$ is the orbital magnetic field, $\phi_0 = h/e$ is the magnetic flux quantum, and $\Theta(x)$ is the Heaviside step function. The model is rescaled using $a \mapsto \tilde{a} = sa$ and $t \mapsto \tilde{t} = t/s$ to reduce the computational cost keeping the Fermi velocity $v_F \propto ta$ unchanged [24]. In all transport calculations, we use $t = 2.8\,\text{eV}$, $a = 0.142\,\text{nm}$ [25], $s = 10$, $\Delta = 1.3\,\text{meV}$ in order to match the MoRe pairing potential [16], and we choose $\chi = 50\,\text{nm}$ to match in order of magnitude the electrostatic estimations of a similar system [26]. In this section, we follow [16] and use a superconductor with a square lattice to study the intervalley scattering along a translationally invariant interface. In the later sections we use a lattice-matched superconductor to isolate other sources of chiral state mixing. In band structure calculations, we use a larger value of $\Delta = 0.05t$ in order to demonstrate more clearly the dispersion relation of the Andreev states.

We compute conductance using a 3-terminal device with the NS interface oriented either along armchair or zigzag direction, as shown in Fig. 1(a). We choose the size of the NS interface to be 600 nm – the same as the experiment reported in Ref. [16]. In both cases, the two edges adjacent to the NS interface are zigzag-type in order to mimic the valley-polarizing behavior of a generic lattice termination boundary. The normal leads are also in the quantum Hall regime to both keep a low computational cost and ensure the lack of extra scattering processes. The distance between the NS interface and the leads, as well as the width of the normal leads is $250\,\text{nm} \gg l_B$. The conductance between the two normal leads equals

$$G_{xx} = \frac{e^2}{h} \sum_{i,j} \left(T_{ee}^{ij} - T_{he}^{ij}\right), \tag{6}$$

where $T_{ee}^{ij}$ is the probability of an electron from channel $j$ in the source lead $N_1$ to transmit as an electron to the channel $i$ in the drain lead $N_2$, and $T_{he}^{ij}$ is the probability of an electron to transmit as a hole.

We show the band structures of the NS interface and the corresponding conductances in Fig. 2. If the NS boundary is parallel to the armchair orientation [Fig. 2(a)], the conductance

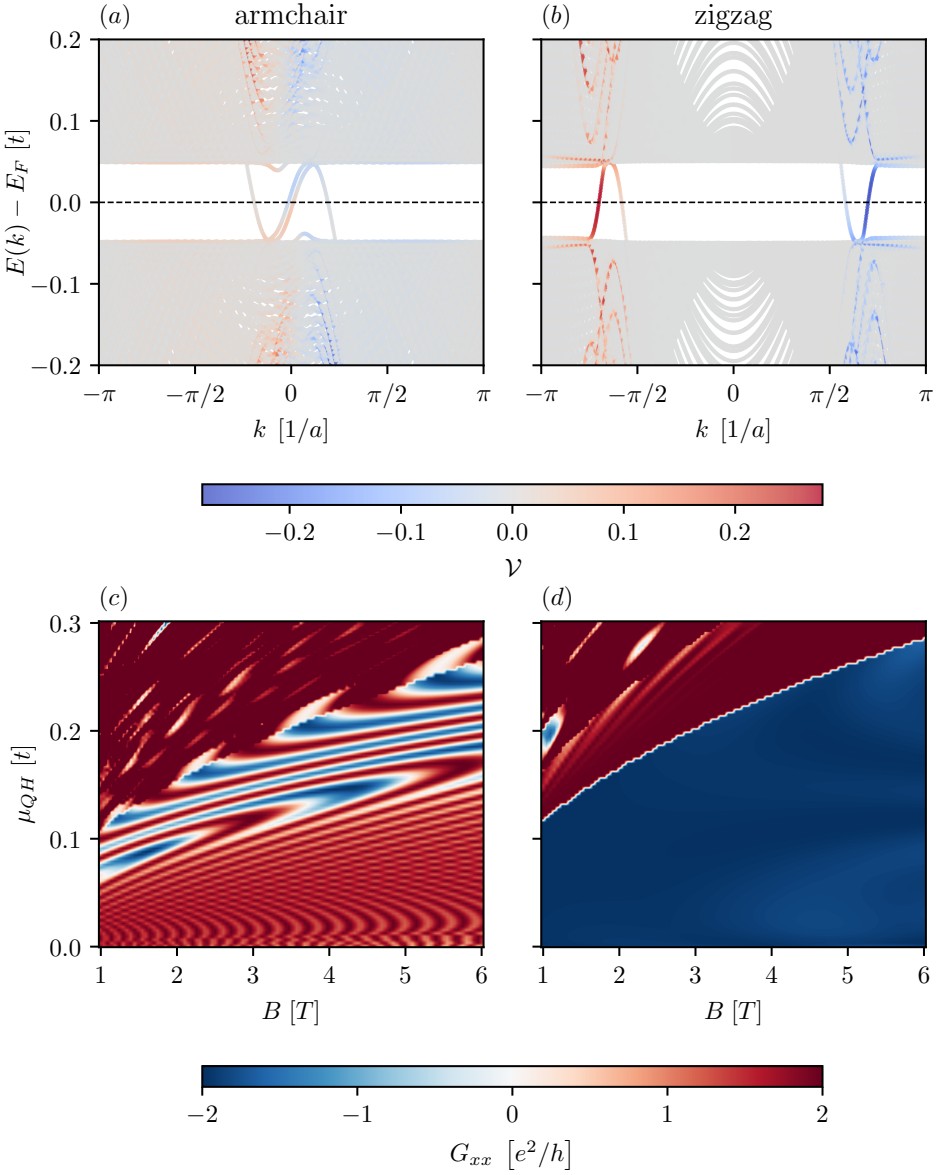

Figure 2: Band structure of a NS ribbon with (a) armchair and (b) zigzag terminations with $\mu_{QH} = \mu_{SC} = 0.05t$, $B = 1T$. The colors indicate the valley expectation value at the NS interface, defined in Eq. 4, with $\chi = 5$. We see that the positive velocity modes (chiral Andreev states) occur near $k = 0$ with armchair interfaces and are well-separated in momentum space for zigzag orientation. The resulting nonlocal conductance presents an interference pattern for armchair interfaces (c) while the expected constant value from Eq. 1 is obtained for zigzag interfaces (d).

shows an interference pattern [Fig. 2(c)]. Chiral modes at zigzag interfaces, on the other hand [Fig. 2(b)], have constant nonlocal conductance [Fig. 2(d)], in agreement with the analytical model. We therefore confirm that the interference of chiral Andreev states at an ideal interface is highly sensitive to its orientation. Moreover, in the generic case we expect the NS conductance to be nearly constant [19]. Since the experiment [16] did not control the lattice orientation down to the required 1° precision, we turn to alternative phenomena that can explain the observed oscillations of the downstream resistance.

## 3 Effect of disorder

To increase the transition rate between the two chiral states, we add short-range disorder modeled as a uniformly-distributed uncorrelated onsite potential:

$$\mathcal{H}_{\text{disorder}} = \sum_{i \in \text{edge}} \psi_i^\dagger \delta\mu_i^{(\text{edge})} \tau_z \psi_i + \sum_{i \in \text{SC}} \psi_i^\dagger \delta\mu_i^{(\text{SC})} \tau_z \psi_i, \tag{7}$$

with $\delta\mu_i^{(\text{edge})}, \delta\mu_i^{(\text{SC})} \in [-t, t]$, illustrated in Fig. 3(a). We consider two relevant types of disorder. To simulate the disorder at the edge of graphene, we apply the disorder potential ($\delta\mu_i^{(\text{edge})}$) within 6 nm from the graphene edges adjacent to the NS interface. Because this disorder extends into the bulk over a length smaller than the magnetic length $l_B = \sqrt{\hbar/eB}$, it does not introduce additional conduction channels, but it breaks the valley polarization of the edge states. Furthermore, in order to approximate the effect of coupling to a strongly disordered MoRe superconductor [16], we add $\delta\mu_i^{(\text{SC})}$ in the superconduting region ($x > 0$).

We observe that conductance in the presence of edge disorder varies only slowly [Fig. 3(c)] because the edge states near the NS interface still maintain a definite valley isospin [Fig. 3(b)], and the propagation along the NS interface preserves valley number. On the other hand, scattering caused by disorder in the superconducting region leads to strong irregular oscillations (Eq. 1) at all values of $\mu_{QH}$ and $B$ [Fig. 3(c)]. We observed that the quantitative behavior is unchanged when a different random seed was used to generate the disorder landscape.

## 4 Effects of Fermi level mismatch

Due to a work function difference, the superconductor dopes the graphene layer near the NS interface. According to electrostatic simulations, the region with increased charge carrier density extends over tens of nanometers [26]. However, due to its smooth potential profile, it preserves valley polarization. The potential mismatch does not qualitatively change most of the proximity physics in graphene, and therefore the potential profile is frequently approximated by a step function, as done *e. g.* in Refs. [6, 27].

We observe that in quantum Hall regime, the doping by the superconducting contact increases the filling factor near the NS interface. When $\chi \gtrsim l_B$ and $\mu_{SC} > \hbar v/l_B$, the smooth potential step introduces additional counter-propagating bands at the Fermi energy, shown in Figs. 4(a), and therefore Eq. 2 does not hold. Because the edge states at the normal edge couple with the bands at the NS interface, including the additional counter-propagating ones, valley conservation no longer constrains the conductance to the quantized value. In agreement with this expectation, including the doping by the superconductor in the numerical simulations produces an interference pattern in the nonlocal Andreev conductance, shown in Fig. 4(b). Naturally, interface disorder spoils this regular interference pattern due to the interface doping.

## 5 Experimental relevance

Comparing the alternative mechanisms, we observe that irregular oscillations, similar to the ones observed experiment, only arise due to the NS interface disorder. Thus, irregular sign changes in nonlocal transport measurements are an indication of a disordered NS interface. On the other hand, devices with a clean NS interface, are expected to have a conductance that is either nearly constant or that exhibits regular oscillations caused by the Fermi level mismatch. While our analysis disregarded the effect of Bogoliubov vortices, we expect that these act as

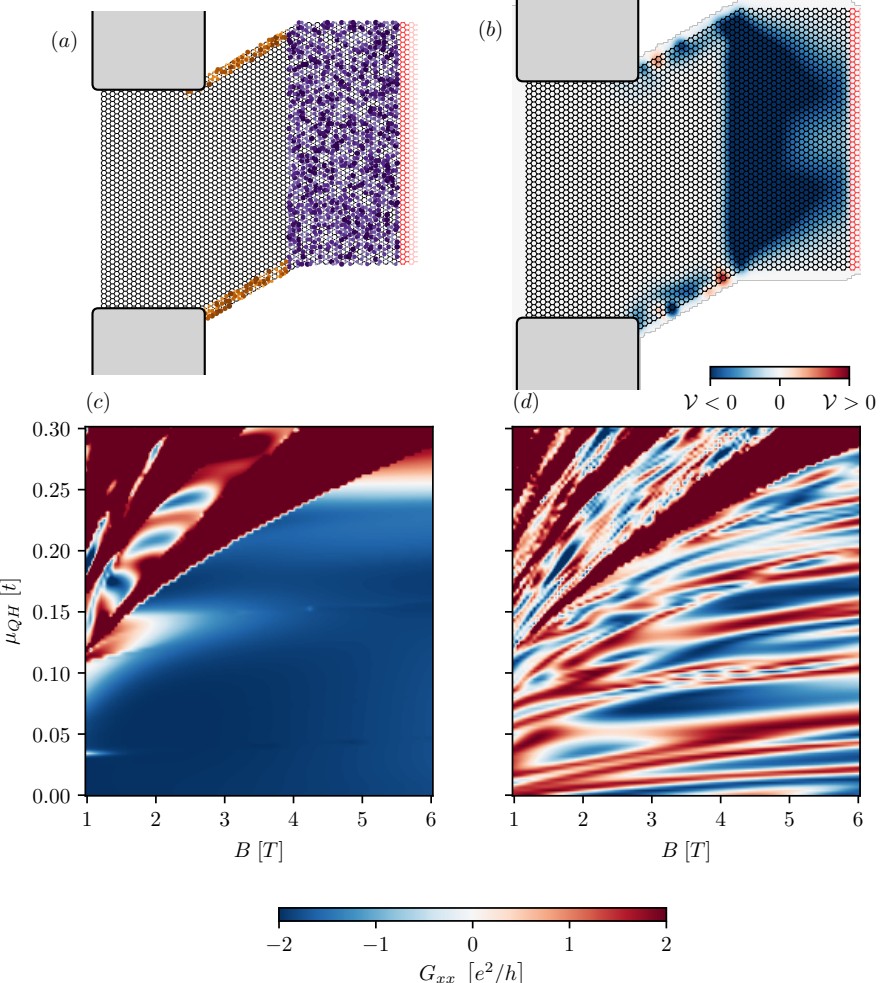

Figure 3: Effects of disorder on the downstream resistance. (a) Scheme of disorder landscape: orange represents $\delta\mu_i^{(\text{edge})}$; purple represents $\delta\mu_i^{(\text{SC})}$. (b) Valley polarization of the edge states with finite $\delta\mu_i^{(\text{edge})}$. One can observe that valley number is still nearly conserved. The change in the nonlocal conductance with finite $\delta\mu_i^{(\text{edge})}$ (c) is minor and the conductance is still nearly constant in a large area of the parameter space. When $\delta\mu_i^{(\text{SC})}$ is finite (d), however, the effects are much stronger and persistent for all $\mu$ and $B$. The other parameters are choosen to be the same as in Fig. 2.

quasiparticle sinks and lower the overall magnitude of the nonlocal conductance, similar to what is observed in Ref. [16] (see the detailed discussion in App. C).

Recent experimental works [15–17] motivate the combination of quantum Hall effect and superconducting order in graphene as a platform for Majorana zero modes [28]. Our results cover two different situations that prevent the existence of a non-trivial topological phase: (i) the strong intervalley scattering caused by disorder closes the topological gap; (ii) Fermi level mismatch promotes the population of undesired edge states along the NS interface [28,29]. Therefore, there is a need to fabricate high-quality graphene/superconductor heterostructures, *e.g.*, substituting the current superconductor deposition techniques by stacking van der Waals materials, such as NbSe$_2$ [26]. Moreover, future works would benefit from electrostatic simulations to properly analyze the edge states population at the NS interface. Alternatively, experiments should explore routes to suppress the Fermi level mismatch.

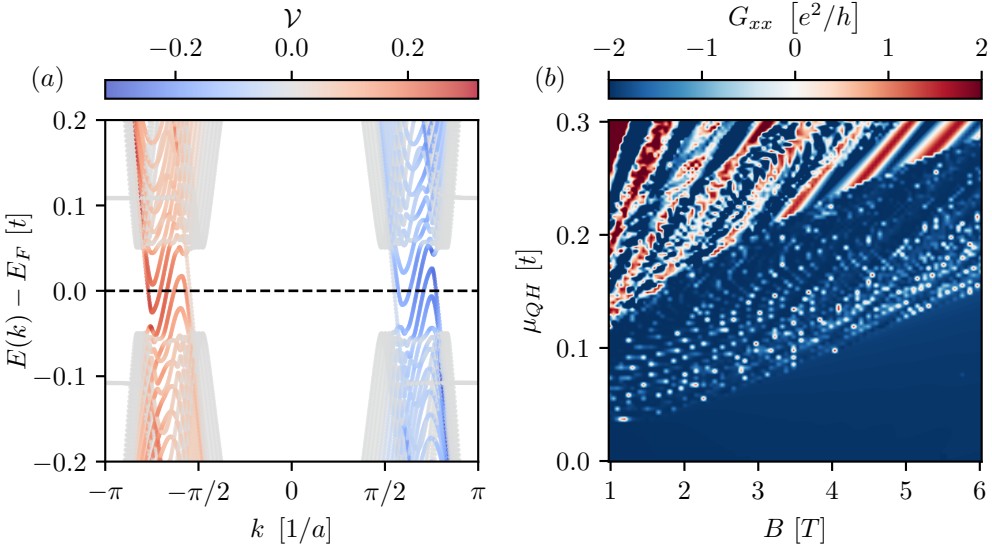

Figure 4: (a) Band structure of a NS ribbon with Fermi energy mismatch ($\mu_{QH} = 0.1t$, $\mu_{SC} = 0.5t$, and $B = 1T$) and corresponding valley expectation value at the NS interface. One can clearly see the presence of additional counter-propagating edge states. (b) Nonlocal conductance of a system with finite Fermi level mismatch ($\mu_{SC} = 0.5t$). There is a clear deviation from the predicted constant value of Eq. 1 due to the extra propagating modes.

## 6 Conclusion

We analysed three mechanisms responsible for fluctuations of Andreev conductance in quantum Hall graphene devices. We concluded that a clean NS interface couples edge states from different valleys only if it is precisely aligned with the armchair direction—an unlikely occurrence in experimental devices. Turning to imperfect interfaces, we observed that short-range disorder enables scattering between the two edge states with opposite valley polarizations, leading to irregular Andreev conductance oscillations that resemble the experimental data [16]. Finally, we found that, even if the NS interface is clean, the Fermi level mismatch populates additional counter-propagating edge states along the NS interface and therefore leads to Andreev conductance oscillations. We believe that our analysis cover all possible Andreev edge state scattering mechanisms, and therefore it allows to qualtitely analyse the NS interface properties. Furthermore, we argue that the intense search for Majorana physics in graphene quantum Hall devices requires improvements in the NS interface quality and proper understanding and control of the Fermi level mismatch.

## Acknowledgements

The authors thank Jose Lado and Kostas Vilkelis for useful discussions.

**Author contributions**   A.R.A. formulated and supervised the project. A.L.R.M., I.M.F., and C.-X.L. performed numerical calculations and identified the possible causes of conductance oscillations. A.L.R.M. developed the analytical model. The authors wrote the manuscript jointly.

**Funding information** This work was supported by grants #2016/10167-8 and #2019/07082-9, São Paulo Research Foundation; by a subsidy for top consortia for knowledge and innovation (TKl toeslag) by the Dutch ministry of economic affairs; and by VIDI grants 680-47-537 and 016.Vidi.189.180.

## Data availability

The code and data used to produce the figures and derive the effective model are available in Ref. [30]. We use Adaptive [31] to efficiently sample $k$-space for the band structure calculations.

## A  Valley-dependence of Andreev reflection in quantum Hall graphene

Andreev reflection is a process in which an incoming charge carrier reflects as its time-reversal partner. In graphene, it means that electrons convert to holes with opposite valley isospin $\nu \otimes \tau$ (electrons and holes from the same valley have opposite isospins) [6, 27]. Thus, charge and valley isospin densities are correlated. Moreover, boundary conditions applied to graphene's lowest Landau level result in valley-polarized edge modes [21, 22]. We can observe both phenomena by computing the local values of valley, valley isospin, and charge densities, as shown in Fig. 5. We compute the valley density as the expectation value of the anti-Haldane operator, $\nu_z$ [32, 33]. In the presence of a magnetic field, we must include a Peierls phase $\phi_{ij}(B)$:

$$\nu_z(B) = \frac{i}{3\sqrt{3}} \sum_{\langle\langle i,j \rangle\rangle} \eta_{ij} s_z^{ij} e^{i\phi_{ij}(B)} c_i^\dagger c_j, \tag{8}$$

where $\langle\langle i,j \rangle\rangle$ denotes a sum performed over the next-nearest-neighbors, $\eta_{ij} = \pm 1$ for a clockwise/anticlockwise hopping, and $s_z = \pm 1$ if $\mathbf{r}_i$ is in the A/B sublattice.

In a two-terminal setup with a NS interface, the longitudinal conductance in the lowest Landau level was previously shown to be [19]:

$$G_{NS} = \frac{2e^2}{h}(1 - \cos \Phi), \tag{9}$$

where $\Phi$ is the angle difference between the valley isospins of the states entering and leaving the superconductor, depicted in Fig. 6(a). It turns out that $\Phi$ depends on the geometry, resulting in constant conductance with different values, as shown in Fig. 6(b).

It is straightforward to compute the nonlocal conductance $G_{xx}$ of a 3-terminal device as the one depicted in Fig. 1. First, we take $\Phi = \pi$. Then, we notice that

$$G_{xx} = \frac{2e^2}{h} - G_{NS} = \frac{2e^2}{h} \cos \Phi, \tag{10}$$

leading to Eq. 1.

## B  Low-energy model derivation

To derive the low-energy effective model presented in Sec. 2, we start with the valley-symmetric Dirac-Bogoliubov-de Gennes Hamiltonian [21] version of Eq. 3 for an infinite system along the



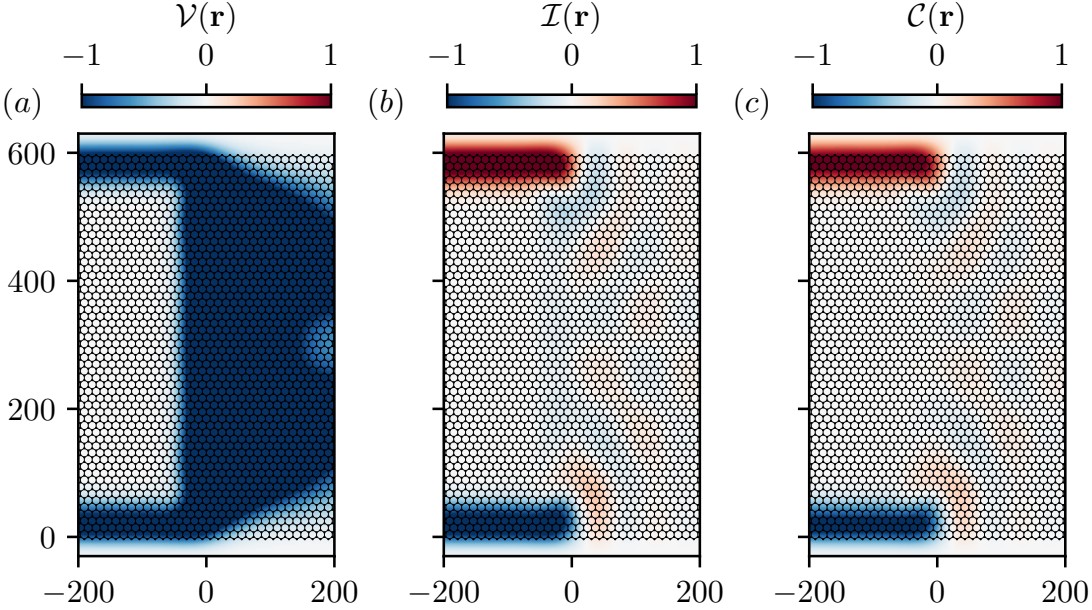

Figure 5: (a) Valley, $\mathcal{V}(\mathbf{r}) = \langle\mathbf{r}|v_z(B)|\mathbf{r}\rangle$, (b) valley isospin, $\mathcal{I}(\mathbf{r}) = \langle\mathbf{r}|\tau_z \otimes v_z(B)|\mathbf{r}\rangle$, and (c) charge, $\mathcal{C}(\mathbf{r}) = \langle\mathbf{r}|-\tau_z|\mathbf{r}\rangle$, densities near a NS interface ($x = 0$) with $B = 1T$ and $\mu = 0.075t$. It is visible that valley number is conserved. Moreover, valley isospin and charge are highly correlated. The length is in units of nm (the honeycomb crystal in the backgound is purely illustrative).

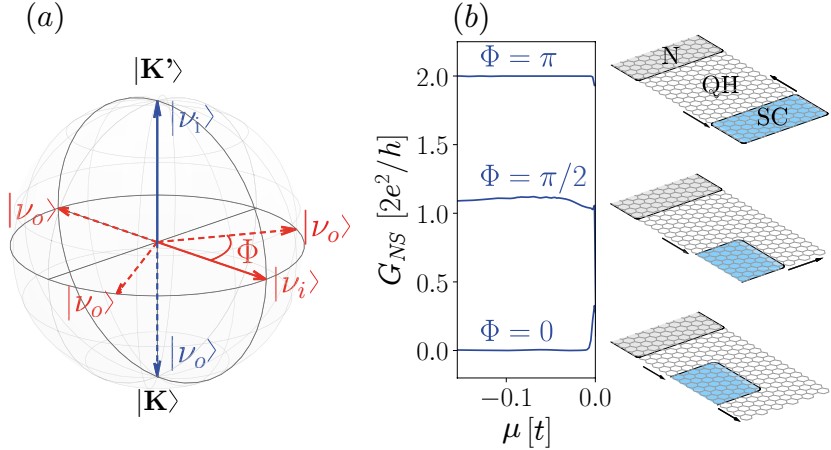

Figure 6: (a) Incoming $v_i$ (solid) and outgoing $v_o$ (dashed) valley isospins on the Bloch sphere. Modes propagating along a zigzag boundary are valley polarized (blue) whereas in an armchair interfaces they are composed by a linear combination of both valley states. (b) $G_{NS}$ conductance plateaus from setups with with different arrangements of the superconductor on a NS junction. The angle difference $\Phi$ between the valley isospin of incoming and outgoing modes depends only on the device geometry.

$y$-axis. We also take $\chi \to 0$, such that $f(\mathbf{r}_i) \to \Theta(\mathbf{r}_i)$. The Hamiltonian reads

$$H = \hbar v \tau_z \otimes \left[-i v_0 s_x \partial_x + (k v_0 + k_0 v_z) \otimes s_y\right] - \mu(x)\tau_z \otimes v_0 \otimes s_0$$
$$+ \tau_0 \otimes v_0 \otimes v\mathbf{A}(x) \cdot \mathbf{s} + \Delta(x)\tau_x \otimes v_0 \otimes s_0 + \omega \tau_0 \otimes v_x \otimes s_0, \qquad (11)$$

where the $\nu_i$ and $s_i$ Pauli matrices act on valley and sublattice spaces, $k$ is the momentum along the $y$-direction, $\pm k_0$ are the Dirac nodes momenta for an arbitrary nanoribbon orientation, amd $\nu$ is the Fermi velocity. We consider a Fermi level mismatch by taking

$$\mu(x) = \begin{cases} \mu_{QH}, \text{ for } x < 0, \\ \mu_{SC}, \text{ for } x > 0. \end{cases} \tag{12}$$

Finally, we allow valley mixing by coupling the two valley states with $\omega$.

We now compute the effective Hamiltonian using first order in perturbation theory [28, 29, 34]. The perturbation is

$$H_{\text{pert}} = \hbar\nu\left(k\rho_0 + k_0\rho_z\right) \otimes s_y + \omega\tau_0 \otimes \nu_x \otimes s_0, \tag{13}$$

and the unperturbed term is

$$H_0 = H - H_{\text{pert}}. \tag{14}$$

Thus, the effective Hamiltonian is obtained by computing

$$(H_{\text{eff}})_{ij} := \langle\psi_i|H_{\text{pert}}|\psi_j\rangle, \tag{15}$$

where $\{\psi_i\}$ are the zero-energy solutions of $H_0$.

In order to find $\{\psi_i\}$, we solve

$$\partial_x\psi = m(x)\psi, \tag{16}$$

$$m(x) = \frac{i\Gamma_1^{-1}}{\hbar\nu}\left[eBx\Theta(-x)\Gamma_4 + \mu(x)\Gamma_0 - \Delta\Theta(x)\Gamma_3\right], \tag{17}$$

where we used $\Gamma$-matrices defined as:

$$\Gamma_0 := \tau_z \otimes \rho_0 \otimes s_0, \tag{18}$$
$$\Gamma_1 := \tau_z \otimes \rho_0 \otimes s_x, \tag{19}$$
$$\Gamma_2 := \tau_z \otimes \rho_0 \otimes s_y, \tag{20}$$
$$\Gamma_3 := \tau_x \otimes \rho_0 \otimes s_0, \tag{21}$$
$$\Gamma_4 := \tau_0 \otimes \rho_0 \otimes s_y, \tag{22}$$
$$\Gamma_5 := \tau_z \otimes \rho_z \otimes s_y. \tag{23}$$

Thus,

$$\psi(x) = \psi_i(x)e^{\lambda_i(x)}, \tag{24}$$

where $\psi_\alpha(x)$ and $\lambda_\alpha(x)$ are the eigenvectors and eigenvalues of

$$M(x) = \int_0^x d\xi \, m(\xi). \tag{25}$$

We then have

$$\psi_i = \frac{1}{\mathcal{N}}\left(\psi_i^{x<0}\Theta(-x) + \psi_i^{x>0}\Theta(x)\right), \quad i = 1, 2, \tag{26}$$

where $\mathcal{N}$ is the normalization constant,

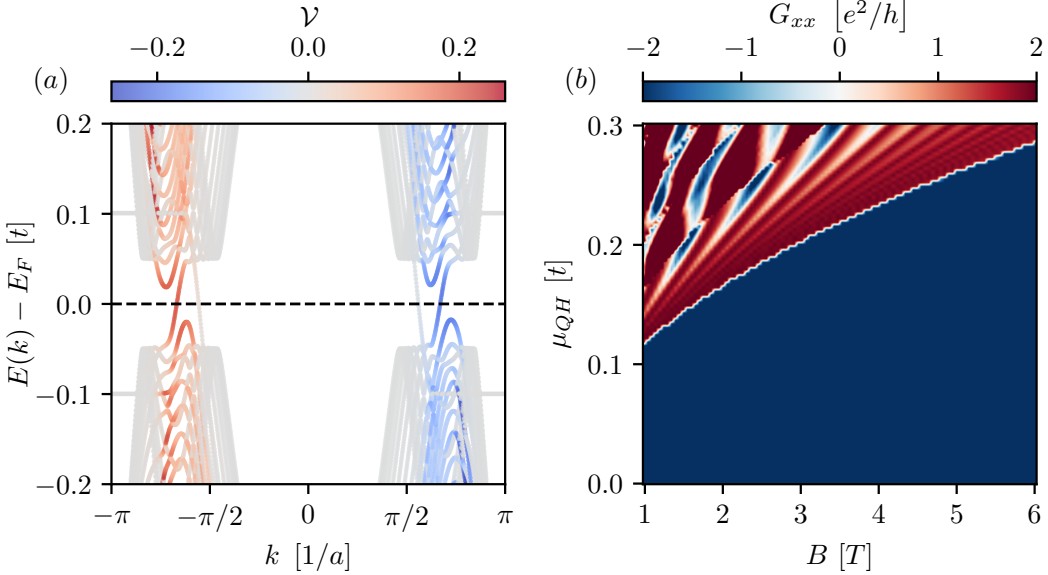

Figure 7: (a) Band structure of a NS ribbon with sharp Fermi energy mismatch ($\mu_{QH} = 0.1t$, $\mu_{SC} = 0.5t$, $B = 1T$, and $\chi = a/1000$) with corresponding valley expectation value. One can see that valley number is still appoximately conserved. (b) Nonlocal conductance of a system with finite Fermi level mismatch for the same parameters. Due to the lack of intervalley mixing, $G_{xx} = -2e^2/h$ at the lowest Landau level.

$$\psi_1^{x<0} = \begin{pmatrix} g(x) \\ i \\ 0 \\ 0 \\ ig(x) \\ 1 \\ 0 \\ 0 \end{pmatrix} \alpha(x), \quad \psi_1^{x>0} = \begin{pmatrix} i \\ i \\ 0 \\ 0 \\ 1 \\ 1 \\ 0 \\ 0 \end{pmatrix} \beta(x), \tag{27}$$

and

$$\psi_2^{x<0} = \begin{pmatrix} 0 \\ 0 \\ g(x) \\ i \\ 0 \\ 0 \\ i \\ g(x) \end{pmatrix} \alpha(x), \quad \psi_2^{x>0} = \begin{pmatrix} 0 \\ 0 \\ i \\ i \\ 0 \\ 0 \\ 1 \\ 1 \end{pmatrix} \beta(x), \tag{28}$$

with

$$g(x) = -\frac{2\mu_{QH}x}{Bex^2 - 2\left(B^2e^2x^2/4 - \mu_{QH}^2\right)^{1/2}|x|}, \tag{29}$$

$$\alpha(x) = e^{-\frac{\left(B^2e^2x^2/4 - \mu_{QH}^2\right)^{1/2}|x|}{\hbar v_F}}, \tag{30}$$

$$\beta(x) = e^{-\frac{x(\Delta - i\mu_{SC})}{\hbar v_F}}. \tag{31}$$

It is easy to see that $\psi_1$ is a linear combination of an electron state at the valley $K$ with a hole state at valley $K'$, while $\psi_2$ is the opposite. Using Eq. 15, it is straightforward to obtain a Hamiltonian with the form of Eq. 2.

The valley mixing has several possible origins. For example, lattice mismatch, sharp electrostatic potential barriers ($\chi \sim a$), and disorder. In the main text, we argue that in clean systems intervalley coupling is negligible for arbitrary lattice orientations. Our arguments are supported by numerical simulations of a system with a square superconducting lattice. In Fig. 7 we show that a sharp electrostatic potential mismatch at the NS interface also results in a negligible intervalley coupling. Namely, valley number is still a good quantum number and downstream conductance is constant.

## C   Absorption of quasi-particle excitations by the superconductor

The quasiparticles absorption by the superconductor reduces the probability of outgoing electrons and holes at the end of the interface. We effectively add a "survival probability" $P_{\text{surv}}$ by modifying the system: we attach a metallic lead to the superconducting region such that quasiparticles can now tunnel through the superconductor with a finite probability. We choose the size width of the superconducting region to be $250\,\text{nm}$, such that it is larger than the superconducting coherence length. Thus, the nonlocal conductance change as [16]

$$\tilde{G}_{xx} = P_{\text{surv}} G_{xx} , \tag{32}$$

where $G_{xx}$ is given by Eq. 10. Thus, the nonlocal conductance is suppressed, as seen in (Fig. 8, following experimental results [16].

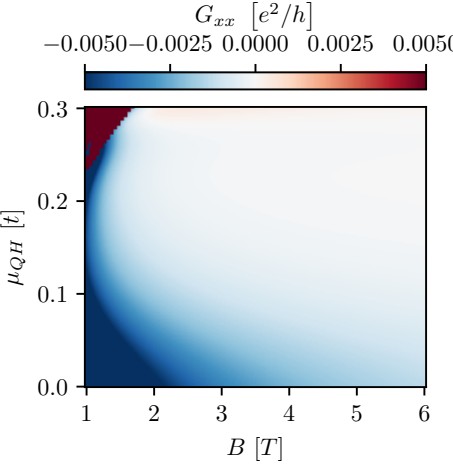

Figure 8: (a) Quasiparticle tunneling to the superconductor suppresses the nonlocal conductance. We choose the same parameters as the ones used in Fig. 2 but with a metallic lead ($\Delta = 0$) connected to the superconductor in the scattering region.

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
