# Peer review of "Mechanisms of Andreev reflection in quantum Hall graphene"

_SciPost Physics Core, doi:SciPost Phys. Core 5, 045 (2022)_

## Round 2 · Referee Report · Anonymous (Referee 1) · 2021-10-29

Report

This work investigates the origin of Andreev interference in a superconductor-graphene heterostructure in the quantum Hall regime. The authors identify three mechanisms leading to deviations from constant conductance. They found that while lattice and Fermi level mismatch at perfect interfaces generate a regular interference pattern in nonlocal conductance, an irregular interference pattern occurs only in the presence of a strong disorder. This latter effect allows them to explain the origin of irregular interference patter in a recent experiment, Ref.[16].

I find the paper very interesting, clearly written, and timely. For these reasons I recommend its acceptance for publication in the current form.
  • validity: -
  • significance: -
  • originality: -
  • clarity: -
  • formatting: -
  • grammar: -

Author:  Antonio Manesco  on 2022-02-16  [id 2210]

(in reply to Report 1 on 2021-10-29)

We thank the referee for their positive evaluation of our work.

---

## Round 2 · Referee Report · Anonymous (Referee 2) · 2021-11-18

Report

This paper addresses the interesting problem of interference between chiral Andreev edge states, the subject of several recent experiments. The main advance made here is to study the effect of disorder. The authors argue that the signal seen in one of the recent experiments [Zhou et al., Ref. 16] is intrinsically due to disorder. The authors are certainly correct that disorder has a large effect on the experimental results, though I am not persuaded that disorder is intrinsically necessary to produce the experimental signal. Nevertheless, I recommend that this work be published by SciPost. Before publication, however, there are a number of issues that should be addressed, as follows.

  1. A key aspect of the authors' argument is that the interface between the graphene and the superconductor is smooth (i.e., in the absence of disorder). The authors should highlight this more clearly in their text to ensure that the reader is fully aware that they are working in this regime. They change the chemical potential smoothly over 50 nm which corresponds to 2-4 magnetic lengths or about 35 tight-binding lattice constants. The authors give only one reference for this 50 nm number--Ref. 26. Since this is such a key part of the argument, it would be good to support this point with more evidence or references, if possible.

  2. The argument in the second paragraph of Sec. 2 is weak. The authors estimate the magnitude of intervalley coupling using $m\sim vk_F$ and use the $k_F$ corresponding to the graphene envelope function--that is, the deviation from the Dirac point. (Note that this is not a value for $k$ in Eq. (2), which denotes the total crystal momentum, and so is somewhat confusing.) But why not use the $k_{F}$ in the superconductor, which is order $1/a$ ? Because the gap in the superconductor is much less than the cyclotron gap in the graphene, most of the quasiparticle wavefunction is in the superconductor, so it seems to me that $k_{F,\text{super}}$ is more appropriate. Using their argument, one immediately concludes $m\sim v k_0$, in contrast to what they say.

  3. When the superconductor is disordered (the most interesting case), are the results sensitive to the magnitude of the disorder? How many different disorder configurations did the authors investigate (approximately)?

  4. Labeling the additional modes in Sec. 4 and Fig. 4 as "non-chiral" is premature. For a mode to be non-chiral, the transverse wavefunction of the mode should be the same for the right-moving and left-moving excitations. So, to make the point that the additional modes are non-chiral, the authors should show that the transverse wavefunction is (approximately) independent of the direction.

In the quantum Hall regime, the longitudinal wavevector of the excitation is directly connected to its transverse position--in the semiclassical regime, the transverse position of the guiding center is proportional to the longitudinal wavevector. In the dispersion in Fig. 4(a), the difference in $k$ between the various modes at the chemical potential seems to be pretty substantial-- $\delta k \,\ell_B$ does not appear to be small. Thus, I would have thought that they are separated by a distance of order $\ell_B$, and so are still chiral modes. Is this not the case?

Since the interference pattern produced by the additional modes shown in Fig. 4 is not very strong, is it possible that the weak interference pattern is due to the modes remaining chiral?

  1. In connection with Fig. 3(b), the authors state that they "observe that conductance in the presence of edge disorder varies only slowly [Fig. 3(b)] because the edge states near the NS interface still maintain a definite valley polarization, and ...". While this seems like a very plausible explanation, the authors haven't actually shown that valley polarization at a disordered edge is maintained. The sentence should either be rephrased as a likely explanation, or the authors should show the valley-density for a disordered edge--it seems that they have this readily available since they show it for a clean case in App. A.

  2. The use of the word "exhaustive" in the conclusion is not warranted and should be removed:
    "We believe that our analysis of the Andreev edge state scattering mechanisms is exhaustive, ...". This is certainly not an exhaustive study; for instance, different disorder strengths have not been studied, and the authors have not varied the smoothness of the interface. There are clearly quite a few ways in which the work could be extended.

  3. The conclusion about Ref. 16, Zhou et al., is, in my view, somewhat misguided. The authors are accurate in their portrayal of the theoretical work in Ref. 16, which presents results only for the clean (not disordered) case. However, the paper is primarily experimental, and the authors make clear that disorder has a large effect on their experimental results. Note, for instance, these two sentences from Zhou et al.:

"Experimentally, the beating pattern between the two CAESs is likely to be affected by multiple parameters, such as the $\textit{interface roughness, disorder potential}$ electron density profile near the contact and even positions of vortices in the superconducting contact (Supplementary Figs. 5 and 6). As a result, the downstream resistance measured as a function of the gate voltage acquires a pattern of $\textit{random but highly reproducible fluctuations}$ (Fig. 1c), in which the signal is positive or negative depending on whether the super-conductor emits predominantly an electron or a hole." (emphasis added)

From the presentation in the current manuscript, a reader would get the impression that there is no awareness of disorder at all in Ref. 16, which is far from the case. I suggest that the authors modify their references to Zhou et al. to correct this imbalance.

  1. A clarification about labeling in two figures is needed. How does the region labeled "N" differ from that labeled "QH" in Figs. 1 and 6? Does this mean that $B=0$ in the region labeled N? If so, have the authors checked that the resistance arising from the change in magnetic field doesn't influence the results?

  2. Throughout the paper, insufficient information is given about the parameters used for the numerical results. First, I did not see information about the geometry. The widths of the leads, length of the interface, width of the disordered region in the superconductor, etc. should be give.

Second, are the parameters used for the conductance in Fig. 4 the same as for Fig. 2?

What is the value of $\Delta$ used in Fig. 5?

I assume the results in Fig. 3 are for a zigzag interface (with zigzag vacuum edges), is that right? In that case the sketch in Fig. 3(a) is highly misleading and should be changed.

In Appendix C, what is the width of the superconducting region before the metallic lead is attached?

  1. In several places there seems to be a typo in that both $B$ and $\Delta$ are nonzero for $x>0$. The authors presumably mean that $B$ is nonzero in the QH graphene while $\Delta$ is nonzero in the superconductor.
  • validity: -
  • significance: -
  • originality: -
  • clarity: -
  • formatting: -
  • grammar: -

Author:  Antonio Manesco  on 2022-02-16  [id 2209]

(in reply to Report 2 on 2021-11-18)

This paper addresses the interesting problem of interference between chiral Andreev edge states, the subject of several recent experiments. The main advance made here is to study the effect of disorder. The authors argue that the signal seen in one of the recent experiments [Zhou et al., Ref. 16] is intrinsically due to disorder. The authors are certainly correct that disorder has a large effect on the experimental results, though I am not persuaded that disorder is intrinsically necessary to produce the experimental signal. Nevertheless, I recommend that this work be published by SciPost. Before publication, however, there are a number of issues that should be addressed, as follows.

We thank the referee for the positive comment and address the queries below. We also updated the manuscript accordingly.

  1. A key aspect of the authors' argument is that the interface between the graphene and the superconductor is smooth (i.e., in the absence of disorder). The authors should highlight this more clearly in their text to ensure that the reader is fully aware that they are working in this regime. They change the chemical potential smoothly over 50 nm which corresponds to 2-4 magnetic lengths or about 35 tight-binding lattice constants. The authors give only one reference for this 50 nm number--Ref. 26. Since this is such a key part of the argument, it would be good to support this point with more evidence or references, if possible.

    We answer the referee's query in two parts.

    A key aspect of the authors' argument is that the interface [...] is smooth[...]. The authors should highlight this more clearly in their text [...]"

    The only important effect of a smooth confining potential is to introduce additional counter-propagating modes, as analyzed in Sec. 4. In the other sections, the smoothness of the potential step does not play an important role. We explain the role of the additional modes in the beginning of the section 4. However, we agree with the referee that the role of the smooth confining potential should be clearly stated, and we have now added a sentence to the second paragraph of Sec. 4, that states the required potential strength and spatial extent.

    The authors give only one reference for this 50 nm number--Ref. 26. Since this is such a key part of the argument, it would be good to support this point with more evidence or references, if possible.

    The exact value of $\chi$ depends on the sample geometry and the material combination. Furthermore, as we explain in the beginning of Sec. 4, this length scale is hard to probe in most transport experiments. It is perhaps due to these difficulties that we were not able to find further references. That said, the point of Sec. 4 is demonstrating a conceptual mechanism, and for that purpose we believe that our discussion is sufficient.

  2. The argument in the second paragraph of Sec. 2 is weak. The authors estimate the magnitude of intervalley coupling using $m \sim v k_F$ and use the $k_F$ corresponding to the graphene envelope function--that is, the deviation from the Dirac point. (Note that this is not a value for $k$ in Eq. (2), which denotes the total crystal momentum, and so is somewhat confusing.) But why not use the $k_F$ in the superconductor, which is order $1/a$ ? Because the gap in the superconductor is much less than the cyclotron gap in the graphene, most of the quasiparticle wavefunction is in the superconductor, so it seems to me that $k_{F,super}$ is more appropriate. Using their argument, one immediately concludes $m \sim vk_0$, in contrast to what they say.

    We thank the referee for noticing this apparent inconsistency. We reformulated our explanation in the manuscript and explain it below as well.

    If we integrate out the superconductor degrees of freedom, the NS interface is reduced to a source of electron-hole and intervalley transitions. Thus, we can estimate an upper bound for intervalley coupling $m$ by the frequency with which quasiparticles in the quantum Hall region hit this wall, which is $\sim 1/v k_{F, gr}$.

    For arbitrary interface orientations, $k_0 \gg k_{F,gr}$. Hence, no intervalley scattering is expected. The exception is when the NS interface is aligned along the armchair orientation, for which $k_0 \sim 0$.

    Our tight-binding simulations also confirm that the intervalley coupling is small:

    • interferece of chiral Andreev edge states is not visible for zigzag interfaces (Fig. 2d);
    • the momentum separation between the two chiral states with an armchair interface is indeed small ($m/v \ll \pi/a$ -- Fig. 2b).
  3. When the superconductor is disordered (the most interesting case), are the results sensitive to the magnitude of the disorder? How many different disorder configurations did the authors investigate (approximately)?

    The computational cost of the simulations did not allow us to make a systematic study of disorder effects. However, we believe that the choice made is physically sound: disorder at the atomic scale should lead to changes in the tight-binding parameters comparable with the magnitude of the hopping constant.

    Although not included in our work, we performed several simulations with smaller disorder amplitude. The qualitative results do not change: even with disorder amplitude about five times smaller than the one used in the simulations, one sees the effects of disorder-induced intervalley scattering. Namely, the oscillations between positive and negative conductance still happen, but the oscillation's period gets larger. One can easily verify the claim above using the code that we made available in Ref. 30.

  4. Labeling the additional modes in Sec. 4 and Fig. 4 as "non-chiral" is premature. For a mode to be non-chiral, the transverse wavefunction of the mode should be the same for the right-moving and left-moving excitations. So, to make the point that the additional modes are non-chiral, the authors should show that the transverse wavefunction is (approximately) independent of the direction.

    In the quantum Hall regime, the longitudinal wavevector of the excitation is directly connected to its transverse position--in the semiclassical regime, the transverse position of the guiding center is proportional to the longitudinal wavevector. In the dispersion in Fig. 4(a), the difference in $k$ between the various modes at the chemical potential seems to be pretty substantial-- $δkℓ_B$ does not appear to be small. Thus, I would have thought that they are separated by a distance of order $ℓ_B$, and so are still chiral modes. Is this not the case?

    We used the term "chiral" here to count single modes moving only in one direction. To avoid an alternative interpretation described by the referee, we replaced "non-chiral" by "additional pairs of counter-propagating modes" in the text. We thank the referee for pointing out this ambiguity of our description.

    Since the interference pattern produced by the additional modes shown in Fig. 4 is not very strong, is it possible that the weak interference pattern is due to the modes remaining chiral?

    Due to the translation invariance along the interface, the left and right movers do not couple. The interference arises from the splitting and recombination of the Andreev edge states at the two ends of the NS interface. The visibility of the interference pattern is therefore governed by the microscopic details of the boundary on either sound of the interface. Additionally, since the modes in a single valley are close in momentum space, a smooth disorder can couple them. That would also increase the observed signal.

  5. In connection with Fig. 3(b), the authors state that they "observe that conductance in the presence of edge disorder varies only slowly [Fig. 3(b)] because the edge states near the NS interface still maintain a definite valley polarization, and ...". While this seems like a very plausible explanation, the authors haven't actually shown that valley polarization at a disordered edge is maintained. The sentence should either be rephrased as a likely explanation, or the authors should show the valley-density for a disordered edge--it seems that they have this readily available since they show it for a clean case in App. A.

    We agree with the referee's statement. We added the corresponding figure to the text.

  6. The use of the word "exhaustive" in the conclusion is not warranted and should be removed: "We believe that our analysis of the Andreev edge state scattering mechanisms is exhaustive, ...". This is certainly not an exhaustive study; for instance, different disorder strengths have not been studied, and the authors have not varied the smoothness of the interface. There are clearly quite a few ways in which the work could be extended.

    There are a lot of systematic studies that could use our results as a starting point. Our work by no means aims to provide quantitative results. Instead, we aim to present the possible mechanisms -- not regimes -- to consider when performing simulations or analyzing experimental data. In that sense, we do believe that our analysis is exhaustive. However, it could be that we still overlooked other mechanisms for this kind of setup. Thus, we rephrase the sentence accordingly.

  7. The conclusion about Ref. 16, Zhou et al., is, in my view, somewhat misguided. The authors are accurate in their portrayal of the theoretical work in Ref. 16, which presents results only for the clean (not disordered) case. However, the paper is primarily experimental, and the authors make clear that disorder has a large effect on their experimental results. Note, for instance, these two sentences from Zhou et al.:

    "Experimentally, the beating pattern between the two CAESs is likely to be affected by multiple parameters, such as the interface roughness, disorder potential electron density profile near the contact and even positions of vortices in the superconducting contact (Supplementary Figs. 5 and 6). As a result, the downstream resistance measured as a function of the gate voltage acquires a pattern of random but highly reproducible fluctuations (Fig. 1c), in which the signal is positive or negative depending on whether the super-conductor emits predominantly an electron or a hole." (emphasis added)

    From the presentation in the current manuscript, a reader would get the impression that there is no awareness of disorder at all in Ref. 16, which is far from the case. I suggest that the authors modify their references to Zhou et al. to correct this imbalance.

    We agree with the referee's statement. By no means do we think the interpretation provided in Ref. 16 is incompatible with our work. Our main intention is to provide other likely mechanisms behind the experimental observations in Ref. 19. In fact, the setup in the simulations performed in Ref. 16 (with armchair interface) helped us to understand that break of valley conservation is required to observe interference of chiral Andreev edge states. We reviewed the manuscript to fix the mentioned imbalance and hope there is no more room for this possible interpretation of our manuscript.

  8. A clarification about labeling in two figures is needed. How does the region labeled "N" differ from that labeled "QH" in Figs. 1 and 6? Does this mean that $B=0$ in the region labeled N? If so,have the authors checked that the resistance arising from the change in magnetic field doesn't influence the results?

    In our simulations, the regions QH and N mean the same thing. We wanted indeed to avoid the effect mentioned by the referee. Moreover, there should be no problem in considering metallic leads, as long as they have a sufficiently large number of modes such that the extra resistance is negligible. The former is more expensive since the S-matrix is larger. So we opted by the first. In our illustration, we distinguish the two regions to emphasize that the leads are semi-infinite and translationally-invariant systems attached to the scattering region.

  9. Throughout the paper, insufficient information is given about the parameters used for the numerical results. First, I did not see information about the geometry. The widths of the leads, length of the interface, width of the disordered region in the superconductor, etc. should be give.

    The details of our simulations are fully available in Ref. 30. We ensured that all systems dimensions were larger than the superconducting coherence length and much larger than $l_B$. The NS interface has 600nm, the same as the experiment. We added this information to the manuscript.

    Second, are the parameters used for the conductance in Fig. 4 the same as for Fig. 2?

    Yes. We specified all the tight-binding parameters used in our simulations below Eq. 6. The same information is also available in Ref. 30.

    What is the value of $Δ$ used in Fig. 5?

    We used $\Delta=1.3meV$, as specified below Eq. 6. The same information is also available in Ref. 30.

    I assume the results in Fig. 3 are for a zigzag interface (with zigzag vacuum edges), is that right? In that case the sketch in Fig. 3(a) is highly misleading and should be changed.

    The sketch was supposed to show the region where we added the onsite disorder and not the exact shape used in the simulations. We, however, agree with the referee that this could be misleading and fixed it.

    In Appendix C, what is the width of the superconducting region before the metallic lead is attached?

    We choose it to be 250nm. Therefore, this dimension is larger than the coherence length and much larger than $l_B$. We now add this information to the manuscript.

  10. In several places there seems to be a typo in that both $B$ and $Δ$ are nonzero for $x>0$. The authors presumably mean that $B$ is nonzero in the QH graphene while $Δ$ is nonzero in the superconductor.

    We thank the referee for spotting the typo. We changed it accordingly.

---

## Round 3 · Referee Report · Anonymous (Referee 2) · 2022-4-30

Report

The authors have addressed several, but not all, of my concerns.

One I would like to insist on, namely point 1 of my first report regarding smoothness of the confining potential. The authors replied, "The only important effect of a smooth confining potential is to introduce additional counter-propagating modes, as analyzed in Sec. 4. ...".

I do not agree. For instance, smoothness is certainly required for the derivation of Eq. (2) given in App. B, where the superconductor is taken to be a honeycomb lattice with the same chemical potential as the graphene. Furthermore, since the spectrum of the two modes in Eq. (2) is degenerate (with respect to $k_0$), Eq. (2) is written for the situation in which the NS coupling is valley isotropic. The usual argument for valley isotropic coupling given in, e.g., Titov and Beenakker relies on a smooth interface. The authors may have other arguments for Eq. (2) in mind but these are not given---the argument written in the paper depends on smoothness.

The authors decided not to implement several of the changes that I brought up in my first report. I think that these would have been beneficial to the reader-- I'm sure I'm not the only reader who was curious about how many disorder configurations were studied, confused by the distinction between the "N" and "QH" in the figures, or had trouble finding the tight-binding parameters, for instance---but I do not feel strongly about them. Likewise, while I still find the authors' qualitative argument for the magnitude of intervalley scattering at the top of p. 4 confusing and suspect, I am willing to give it a pass.

In summary, there is one issue that still requires attention---the smoothness of the potential. Once this is taken care of, the paper will be suitable for publication.
  • validity: -
  • significance: -
  • originality: -
  • clarity: -
  • formatting: -
  • grammar: -

Author:  Antonio Manesco  on 2022-08-03  [id 2708]

(in reply to Report 1 on 2022-04-30)
Category:
answer to question
reply to objection
correction
validation or rederivation
pointer to related literature

We thank the referee for the feedback.

The main remaining concern of the referee is our claim that the main role of the smooth electrostatic potential is limited to introducing additional counter-propagating modes. The referee lists the valley-isotropic superconducting pairing used in the App. B as an additional important effect of the smoothness of the electrostatic potential. The referee then argues that because we use the appendix to support Eq. 2, the said equation therefore also relies on the smoothness of the electrostatic potential.

Furthermore, the referee noticed missing details on how the simulations were performed. We now provide them in the manuscript.

We provide a detailed answer to the report below.

Part 1: Electrostatic potential at the NS interface

One I would like to insist on, namely point 1 of my first report regarding the smoothness of the confining potential. The authors replied, "The only important effect of a smooth confining potential is to introduce additional counter-propagating modes, as analyzed in Sec. 4. ...". I do not agree. For instance, smoothness is certainly required for the derivation of Eq. (2) given in App. B, where the superconductor is taken to be a honeycomb lattice with the same chemical potential as the graphene. Furthermore, since the spectrum of the two modes in Eq. (2) is degenerate (with respect to $k_0$), Eq. (2) is written for the situation in which the NS coupling is valley isotropic. The usual argument for valley isotropic coupling given in, e.g., Titov and Beenakker relies on a smooth interface. The authors may have other arguments for Eq. (2) in mind but these are not given---the argument written in the paper depends on smoothness.

Following the referee's remarks, we realized that the manuscript did not present our argument in a clear fashion. We have used this opportunity to clarify our logic.

The form of Eq. 2 is completely constrained by the particle-hole symmetry. However, in order to relate it to the NS conductance one needs to identify the degrees of freedom in terms of valleys and electrons or holes. Our claim that the two modes are nearly valley-polarized is supported on the one hand by the qualitative argument that we have discussed in the previous review round (the one that the referee is "willing to give it a pass"), and on the other hand by the numerical evidence where we explicitly compute the valley polarization of different eigenstates in presence of strong intervalley scattering due to the lattice mismatch. To fully address the referee's concerns regarding abrupt potential changes we have now also demonstrated the same approximate valley conservation in presence of an atomically sharp potential step (the newly added Fig. 7). The referee, however, is correct in claiming that the App. B relies on the valley-isotropic pairing. The logical relation between the App. B and the Eq. 2 was not clear in the previous version. Specifically, the App. B serves only as an illustration of how Eq. 2 can be derived, rather than as its foundation.

We have rewritten the discussion to make the above points clear. In addition, we have rewritten the discussion in the App. B, and we now include intervalley coupling in the Eq. (12). We improved the presentation of the results by computing the valley polarization of the chiral Andreev states, once again confirming our qualitative argument.

Part 2: Missing information in the manuscript

The authors decided not to implement several of the changes that I brought up in my first report.

I think that these would have been beneficial to the reader—I'm sure I'm not the only reader who was curious about how many disorder configurations were studied

We verified that the qualitative behavior reported is independent on the random seed used to generate the disorder landscape. We now provide in the Zenodo repository the data from one more calculation and modified the code to set the chosen seed as an extra parameter and state in the manuscript that the results are representative. However, we did not perform a sufficient amount of calculations to be able to address universal quantitative behavior. Since this work aims to report possible mechanisms for the experimental observations, a systematic quantitative analysis is beyond the scope of the current manuscript. A quantitative analysis was recently done elsewhere (arXiv:2201.00273).

Figure. Downstream conductance with the same system parameters as Fig. 3 (d) in the manuscript but with a different disorder realization. Image available here.

[I'm not the only reader...] confused by the distinction between the "N" and "QH" in the figures

As long as the contact resistance of the normal leads is negligible, the distinction is irrelevant. There are two possible choices to obtain a vanishing reflection probability: 1. create a semi-infinite metallic lead with a large density of states; 2. create a semi-infinite lead with the same tight-binding description as the scattering region.

Option 1 has the disadvantage of resulting in a large scattering matrix due to a large number of propagating modes. Therefore, in our transport simulations, we choose option 2 to keep a reduced computational cost. We now explicitly state it in the manuscript.

[I'm not the only reader... who] had trouble finding the tight-binding parameters

In the previous response, we added all the missing parameters we could find. We reviewed our manuscript one more time and added system's dimensions that were missing.

Likewise, while I still find the authors' qualitative argument for the magnitude of intervalley scattering at the top of p. 4 confusing and suspect, I am willing to give it a pass.

As outlined in our response in the first part, we have expanded the argument. While we agree with the referee that this argument is only qualitative, we also demonstrate that it fully agrees with the numerical simulations.

---

## Round 3 · Author Response

Dear editor,

We thank the referees for their evaluation. We reviewed our manuscript according to the reports. Hopefully, the implemented changes made our manuscript suitable for publication.

We also want to bring to the referees' and readers' attentions that this work was recently presented as a satellite talk the "Andreev reflection in quantum Hall systems: 2021 state of the union" workshop hosted by the Virtual Science Forum. The video recording includes discussions about the work that might be relevant during the evaluation of the manuscript.

---

## Round 3 · List of Changes

Explicitly wrote the conditions to observe the phenomenon described in Sec. 4. Reformulated the justification for a small coupling between chiral edge states in the clean limit. Added a figure panel to show approximate valley conservation in disorder boundaries. Added further details about our simulations.

---

## Round 4 · Author Response

We thank the referee for the feedback.

The main remaining concern of the referee is our claim that the main role of the smooth electrostatic potential is limited to introducing additional counter-propagating modes. The referee lists the valley-isotropic superconducting pairing used in the App. B as an additional important effect of the smoothness of the electrostatic potential. The referee then argues that because we use the appendix to support Eq. 2, the said equation therefore also relies on the smoothness of the electrostatic potential.

Furthermore, the referee noticed missing details on how the simulations were performed. We now provide them in the manuscript.

We provide a detailed answer to the report below.

Part 1: Electrostatic potential at the NS interface

One I would like to insist on, namely point 1 of my first report regarding smoothness of the confining potential. The authors replied, "The only important effect of a smooth confining potential is to introduce additional counter-propagating modes, as analyzed in Sec. 4. ...". I do not agree. For instance, smoothness is certainly required for the derivation of Eq. (2) given in App. B, where the superconductor is taken to be a honeycomb lattice with the same chemical potential as the graphene. Furthermore, since the spectrum of the two modes in Eq. (2) is degenerate (with respect to $k_0$), Eq. (2) is written for the situation in which the NS coupling is valley isotropic. The usual argument for valley isotropic coupling given in, e.g., Titov and Beenakker relies on a smooth interface. The authors may have other arguments for Eq. (2) in mind but these are not given---the argument written in the paper depends on smoothness.

Following the referee's remarks we have realized that the manuscript did not present our argument in a clear fashion. We have used this opportunity to clarify our logic.

The form of Eq. 2 is completely constrained by the particle-hole symmetry. However, in order to relate it to the NS conductance one needs to identify the degrees of freedom in terms of valleys and electrons or holes. Our claim that the two modes are nearly valley-polarized is supported on the one hand by the qualitative argument that we have discussed in the previous review round (the one that the referee is "willing to give it a pass"), and on the other hand by the numerical evidence where we explicitly compute the valley polarization of different eigenstates in presence of strong intervalley scattering due to the lattice mismatch. To fully addrees the referee concerns regarding abrupt potential changes we have now also demostrated the same approximate valley conservation in presence of an atomically sharp potential step (the newly added Fig. 7). The referee, however, is correct in claiming that the App. B relies on the valley-isotropic pairing. The logical relation between the App. B and the Eq. 2 was not clear in the previous version. Specifically, the App. B serves only as an illustration of how Eq. 2 can be derived, rather than as its foundation.

We have rewritten the discussion to make the above points clear. In addition, we have rewritten the discussion in the App. B, and we now include intervalley coupling in the Eq. (12). We improved the presentation of the results by computing the valley polarization of the chiral Andreev states, once again confirming our qualitative argument.

Part 2: Missing information in the manuscript

The authors decided not to implement several of the changes that I brought up in my first report.

I think that these would have been beneficial to the reader—I'm sure I'm not the only reader who was curious about how many disorder configurations were studied

We verified that the qualitative behavior reported is independent on the random seed used to generate the disorder landscape. We now provide in the Zenodo repository the data from one more calculation and modified the code to set the chosen seed as an extra parameter and state in the manuscript that the results are representative. However, we did not perform a sufficient amount of calculations to be able to address universal quantitative behavior. Since this work aims to report possible mechanisms for the experimental observations, a systematic quantitative analysis is beyond the scope of the current mansucript. A quantitative analysis was recently done elsewhere (arXiv:2201.00273).

Figure 1. Downstream conductance with the same system parameters as Fig. 3 (d) in the manuscript but with a different disorder realization. Image available here.

[I'm not the only reader...] confused by the distinction between the "N" and "QH" in the figures

As long as the contact resistance of the normal leads is negligible, the distinction is irrelevant. There are two possible choices to obtain a vanishing reflection probability: 1. create a semi-infinite metallic lead with large density of states; 2. create a semi-infinite lead with the same tight-binding description as the scattering region.

Option 1 has the disadvantage of resulting in a large scattering matrix due to the large number of propagating modes. Therefore, in our transport simulations, we choose option 2 to keep a reduced computational cost. We now explicitly state it in the manuscript.

[I'm not the only reader... who] had trouble finding the tight-binding parameters

In the previous response we added all the missing parameters we could find. We reviewed our manuscript one more time and added system's dimensions that were missing.

Likewise, while I still find the authors' qualitative argument for the magnitude of intervalley scattering at the top of p. 4 confusing and suspect, I am willing to give it a pass.

As outlined in our response in the first part, we have expanded the argument. While we agree with the referee that this argument only qualitative, we also demonstrate that it fully agrees with the numerical simulations.

---

## Round 4 · List of Changes

• Computed the corresponding valley expectation value in the bandstructure calculations.
  • Added Fig. 7 with bandstructure and downstream conductance of a system interface with sharp electrostatic potential.
  • Clarified the logic to obtain the effective Hamiltonian in Eq. 2.
  • Clarified the logic in App. B and emphasized that the Appendix serves only as an illustration of how Eq. 2 can be derived, rather than as its foundation.
  • Refactored the code to the user to generate multiple disorder realizations and added an illustrative case in the reply to the referee.
  • Added missing parameters in the manuscript.

The changes in the manuscript are highlighted in the PDF available here: https://surfdrive.surf.nl/files/index.php/s/9H0b4Ytaf5rfypI

---

## Editorial Decision

published